# Patterns of Segmental Strain of the Left Ventricle in Extremely Premature Infants

**DOI:** 10.3390/pediatric17060126

**Published:** 2025-12-01

**Authors:** Tatiana Chumarnaya, Evgeniya Gusarova, Natalya Kosovtsova, Svetlana Koltashova, Olga Solovyova

**Affiliations:** 1Institute of Natural Sciences and Mathematics, Ural Federal University, 620002 Yekaterinburg, Russia; chumarnaya@gmail.com (T.C.); soloveva.olga@urfu.ru (O.S.); 2Institute of Immunology and Physiology, Ural Branch of the Russian Academy of Sciences, 620016 Yekaterinburg, Russia; 3Mother and Child Care Research Institute, 620142 Yekaterinburg, Russia; 4Regional Perinatal Center, 454052 Cheiyabinsk, Russia; 5Medical Center Novi, 620089 Ekaterinburg, Russia

**Keywords:** speckle-tracking echocardiography, segmental strain, premature infants

## Abstract

Extremely premature newborns are predisposed to cardiovascular complications due to a number of factors, including myocardial immaturity, hemodynamic changes, and iatrogenic effects. There are few studies on myocardial strain in extremely premature infants during the early neonatal period. The objective of study was to assess the left ventricular (LV) segmental strain in extremely premature newborns during the early neonatal period by employing speckle-tracking echocardiography (STE). This prospective study examined 65 newborns with no signs of hemodynamic impairment during the first 72 h of life. The cohort had a range of birth weights (600–1500 g) and gestational ages (24–35 weeks). The peak strain in 18 LV segments during systole (peak S and time to peak S), and throughout the cardiac cycle (peak G and time to peak G), and during early systolic pre-stretch (peak P and time to peak P) were assessed in the longitudinal, circumferential, and radial directions. We obtained percentile tables of segmental strain characteristics in the longitudinal, circumferential, and radial directions. No dependence of segmental strain on the birth weight, gestational age, or arterial duct closure was found. A positive gradient of the longitudinal strain magnitude was observed from the base to the apex. The highest circumferential and radial strain were observed in LV septum. This study is the first to register and compare the longitudinal, circumferential, and radial LV strain using STE in extremely premature infants with no signs of hemodynamic disturbances during the first 72 h of life. Reference values for segmental strain were established.

## 1. Introduction

Assessment of global cardiac function alone is not always sufficient for the diagnosis and prognosis of heart disease. Numerous studies in adult populations demonstrate that the segmental characteristics of left ventricular myocardial deformation serve as predictors of pathological changes in preserved global cardiac function [1,2].

The biplane Simpson method of assessing global LV function results in high inter- and intra-operator variability and low reproducibility. A more accurate method of assessing LV systolic function is global longitudinal strain, which enables minor changes to be identified at an earlier stage [3]. It is suggested that segmental strain testing may help in detecting focal lesions of the LV myocardium, such as those associated with neonatal coronary pathology (coronary artery origin anomalies, thrombosis, fibroelastosis, etc.). In the presence of segmental pathology, the function of the affected segment may be compensated for by the increased function of healthy adjacent segments, and global systolic dysfunction indices may be within normal limits [2].

Speckle-tracking echocardiography (STE) is a contemporary ultrasound technique used for evaluating myocardial function. This method is based on the analysis of the spatial displacement of speckles reproduced in a grayscale image, which arise as a result of the interaction of the ultrasound beam with myocardial fibers.

Speckles represent ultrasonic imprints that are tracked throughout the cardiac cycle using specialized software. This software facilitates the calculation of key contraction parameters, including deformation, deformation velocity of selected myocardial regions, rotation, twisting, and untwisting [4].

The STE method is a clinical practice that utilizes three primary characteristics to assess cardiac function. Longitudinal strain provides an assessment of the contraction of muscle fibers that extend longitudinally along the heart’s axis, spanning from the base to the apex. Circumferential strain offers an assessment of the contraction of fibers that envelop the heart’s circumference. Radial strain reflects thickening (positive strain) of the heart muscle wall.

The utilization of STE facilitates the examination of the characteristics of segment deformation and the variety of LV torsion mechanics, including strain, rotation, and twisting during postnatal growth and development [4]. Premature newborns, particularly those with extremely low birth weight, are predisposed to cardiovascular complications due to myocardial immaturity, hemodynamic changes, and iatrogenic effects [5]. STE offers the potential for early detection of myocardial dysfunction, which can enhance survival and reduce disability in this patient population.

Studies of myocardial deformation using the STE method in premature infants in the early neonatal period are few in number. The existing studies have focused on the evaluation of global characteristics of right ventricular (RV) and left ventricular (LV) deformation, with a particular emphasis on global longitudinal strain (GLS) [5,6,7,8,9].

To date, there is no detailed description in the world literature of regional, longitudinal, circumferential, and radial strain of the heart chambers in extremely premature infants. The aim of this study is to evaluate regional LV strain in the longitudinal, circumferential, and radial directions in extremely premature newborns during the early neonatal period. In particular, to obtain reference values for segmental LV strain characteristics for the patient group under consideration.

## 2. Materials and Methods

### 2.1. Population

A prospective study was conducted over a period of 1.5 years in the neonatal intensive care unit of the Mother and Child Care Research Institute. The cohort comprised 65 newborns with a birth weight ranging from 600 to 1500 g, gestational age ranging from 24 to 35 weeks. The infants showed no signs of hemodynamic impairment, as assessed by blood pressure, heart rate within gestational reference values, and the absence of a need for cardiotonic or vasopressor support. Newborns with congenital malformations, episodes of hemodynamic impairment, or a heart rate below 120 beats per minute or above 180 beats per minute were excluded from the study. Two infants were excluded from the group due to suboptimal cardiac imaging. The examination was carried out while the newborns were sleeping or in a calm state; sedation was not used. The study was conducted in accordance with the Declaration of Helsinki, and the protocol was approved by the Ethics Committee of Federal State Budgetary Institution ‘Ural Research Institute of Mather and Child Care’ of the Ministry of Health and Social Development of the Russian Federation No. 16 on 6 September 2023.

### 2.2. STE

Echocardiography was performed using a portable ultrasound device (GE Medical Systems (China) Co., Ltd., Wuxi, Jiangsu, China) equipped with a sector phased transducer (12S-RS, 4.5–12 MHz). Cine loops were obtained in B-mode with synchronous recording of ECG of the following sections of the heart: apical-4-chamber (4CH), 2-chamber (2CH), and along the long axis of the left ventricle (APLAX); parasternal at the level of the mitral valve (MV), papillary muscles (PMs), and apex (AP) of the LV.

The frame rate in recording the cardiac cycle is 100–140 frames per second. The images were saved in DICOM format and subsequently analyzed using the EchoPac v. 204. The optimal frame rate was achieved by reducing the sector width and scan depth. This was sufficient even for such a high heart rate. The shallow location and small size of the neonatal heart create favorable conditions for obtaining high-quality images.

Images were acquired with the direction angle most parallel to the region of interest (ventricular walls), in accordance with speckle tracking recommendations, owing to the improved spatial resolution along the ultrasound beam. The contouring zone was superimposed using the reference point placement method. To achieve more accurate results, the contouring area was carefully checked, and manual corrections were made if necessary. Tracking quality was assessed automatically by the software. If inadequate tracking was detected, additional manual corrections were performed. Endocardial tracking quality scoring was not performed.

The image analysis was performed by a single researcher, E.G. The standard echocardiographic parameters of the LV were assessed, including end-diastolic size, end-systolic size, thickness of the interventricular septum, posterior wall, ejection fraction according to Teicholz, cardiac output, and cardiac index [10]. The strain values were also calculated from the STE data (Figure 1).

Peak S of longitudinal or circumferential strain was defined as the maximum negative value during the systole. Peak S is defined on the deformation curve in the systole before the closure of the aortic valve. It reflects the maximum shortening of the segment in systole and is used to assess the contribution of the segment to the pumping function of the heart.

Peak G was defined as the maximum absolute value of longitudinal, circumferential, and radial deformation throughout the cardiac cycle. Peak G can occur either before or after aortic valve closure. If the time position of the peak G is before aortic valve closure, the values of the peak S and peak G coincide. A peak G positioned after aortic valve closure suggests a pattern of post-systolic shortening. Post-systolic shortening may be classified as physiological if its amplitude and duration are small. It is believed that the presence of this pattern with large amplitude and duration indicates an enhancement of myocardial stiffness at the beginning of diastole, including the period of isovolumetric relaxation.

Peak P of longitudinal or circumferential strain was defined as the positive systolic strain peak. If there is no positive peak, it is set equal to 0. Peak P occurs at the beginning of systole, including the phase of isovolumetric contraction with closed valves. Peak P indicates early systolic lengthening of the segment before the onset of its shortening during systole. Normally, it may be observed in normal segments having a small amplitude and duration. In pathology, the duration and amplitude of the P peak may increase and occupy the entire systole indicating a significant damage of the myocardial segment [11].

Figure 2 shows the LV segments for which regional deformations were analyzed and the peak S, G, and P were assessed. Longitudinal strain (SL) characteristics include the following indexes:
Global longitudinal strain (GLS);
-Regional SL Peak S and the time of its achievement (time to peak);-Regional SL Peak G and the time to peak G;-Regional SL Peak P and the time to peak P.Circumferential strain (SC) parameters include the following:
-The regional SC Peak S and the time of its achievement-The regional SC Peak G and the time of its achievement-The regional SC Peak P and the time of its achievementRadial strain (SR) parameters include the regional SR Peak G and the time of its achievement.

The longitudinal and circumferential myocardial strain values were defined by the software as negative percentages. The terms “strain decrease” and “strain increase” were employed to denote the behavior of the strain when approaching and exceeding negative values, respectively. The radial strain values were defined by the software as positive percentages. For the purpose of this study, the terms “strain decrease” and “strain increase” were used to describe the behavior of the radial strain values as they vary from small to large values.

### 2.3. Statistics

The R software environment (RStudio 2025.05.1) was used for statistical analysis. The quantitative characteristics enumerated in Table 1 were described by the median (50), 25, and 75 percentiles. Qualitative variables were characterized by the absolute number and frequency of occurrence in percentages.

The distributions of segmental strain peaks and the time to those peaks were presented by percentile curves. In this particular context, the 5th percentile, 25th percentile, 50th percentile, 75th percentile, and 95th percentile were utilized. Percentiles were chosen to represent the data because this approach provides a robust and intuitive understanding of its distribution. Percentiles do not make any assumptions about the distribution of the data and simply describe it as it is.

The correlations of global and segmental strain characteristics with birth weight and gestational age were assessed using Spearman’s rank correlation with FDR (false discovery rate) correction for multiple comparisons.

We assessed differences in subgroups with open and closed arterial ducts using permutation tests with FDR correction for multiple comparisons. The first, a permutation test was performed for the entire set of segments for each characteristic (Peak G, Peak S, Peak P, Time to Peak G, Time to Peak S, and Time to Peak P). If such a test was significant, then a test was performed for each segment with FDR correction for multiple comparisons.

We used the Wilcoxon test with FDR correction for multiple comparisons to assess statistical differences in the same characteristics (peak G, peak S, peak P, time to peak G, time to peak S, and time to peak P) across different segments and also for comparison longitudinal and circumferential strain peaks.

The significance level at which the null hypothesis was rejected was taken as 0.05.

Intra-observer variability was assessed for GLS in a randomly selected subset of 20 patients. The same observer EG, who is cardiologist with 5 years of experience in speckle-tracking, repeated the measurements on two separate occasions, blinded to the results of the first analysis and to patient identifiers. The time interval between measurements was 1 month. Variability was determined using the intraclass correlation coefficient (ICC) for consistency, Bland–Altman analysis (reporting bias and limits of agreement), and the coefficient of variation (CV). Intra-observer variability: ICC was 0.96 (95% CI: 0.91–0.98). Bland–Altman analysis showed a mean bias of −0.1% with limits of agreement ranging from −1.5% to +1.3%. The coefficient of variation was 4.2%.The analysis demonstrates high intra-observer reproducibility.

## 3. Results

In total, we evaluated 254 global and segmental parameters of longitudinal, circumferential, and radial left ventricular (LV) strain, as determined by the speckle tracking echocardiography (STE) method, in 65 extremely premature newborns without signs of hemodynamic disturbances during the first 72 h of life. The median birth weight in the cohort was 1210 (980; 1420) grams, the median gestational age was 29.7 (27.4; 31.1) weeks. Table 1 presents the description and characteristics of the standard echocardiographic protocol for this group of infants.

We investigated statistical differences in segmental strain characteristics among subgroups weighing between 500 and 999 g (extremely low birth weight, *n* = 21) and between 1000 and 1499 g (very low birth weight, *n* = 44). No statistically significant differences were found. We also performed a correlation analysis between birth weight, gestational age, and the strain characteristics under consideration.

We did not find any statistically significant dependencies between the considered strain peaks, the time at which they were achieved, and gestational age or birth weight (Appendix A). We analyzed differences in the segmental parameters of longitudinal, circumferential and radial LV strain in infants with an open (*n* = 46) and closed (*n* = 19) arterial duct. No statistically significant differences were found (Appendix A). Therefore, we presented data for the entire group of infants without separation by weight, gestational age, and regardless of duct closure.

For longitudinal strain, GLS was calculated by the software using three apical projections (4CH, 2CH, and APLAX), and its median value was −16.4 (−14.5; −18.4). The segmental characteristics of the peak S of longitudinal strain were compared with this value. The available software does not allow for the calculation of global characteristics for circular and radial strain. For the G and P longitudinal peak and all the circular and radial peak times, the medians of all LV segments were used as global characteristics (Table 1).

### 3.1. Longitudinal Strain

Figure 2, Figure 3 and Figure 4 show the distributions of peaks S, G, P and time to peaks relative to the location of LV segments for longitudinal strain. Percentile tables of all considered parameters corresponding to these figures are provided in the Appendix A.

The peak strain of the left ventricular (LV) segments during systole was estimated by the peak S value. In the considered cohort, a gradient of Peak S values was observed from the apex to the base of the LV (heatmap in Appendix A). Thus, statistically significantly larger absolute values of Peak S were observed in most apical segments compared to the middle and base segments. Segmental values of peak S longitudinal strain were compared with the GLS. Significant differences were observed in the basal and middle segments, where the strain (absolute peak S value) was statistically lower than the absolute value of GLS. At the same time, the strain values of the apical level of the anterio-septal, septal, and inferior walls were significantly higher (Appendix A). This reveals significant quantitative differences in the contribution of different LV segments to LV longitudinal systolic strain.

The time to peak S in the segments did not differ statistically and did not differ from the global median time to peak S for all segments (Appendix A). For most segments of the LV, no significant correlation was found between peak S and the time to peak S (Appendix A).

The P peak characterized the early systolic elongation of the segments. The peak P (Figure 4 top) was greater than zero in the basal segments of the lateral, posterior, and inferior walls, the middle level of the lateral and posterior walls, and the lateral at the apical level. Compared with the global median peak P, significantly larger peaks were observed in the lateral wall, the posterior wall at the basal and middle levels, and the apical segment of the septum. Significantly smaller peak values were observed in the middle segment of the anterio-septal wall and the apical segment of the posterior wall (Appendix A).

Segmental times to peak P compared to the global median time to peak P (Figure 4 bottom) were significantly longer in the basal parts of the anterior and lateral walls and in all segments of the septum (Appendix A). For all segments, a significant positive correlation was established between the peak P value and the time it was reached (Appendix A). This correlation means the greater the positive strain of the segment, the longer time it takes to reach the peak and the longer the entire phase of segmental pre-stretch during the systole.

For most segments of the LV, a significant positive correlation was found between the P peak and the S peak. At the same time, the time to peak P was positively correlated with the time to peak S (Appendix A).

For all LV segments, a significant positive correlation was established between Peak S and Peak G, Time to Peak S and Time to Peak G (Appendix A).

Similarly to the peak S, there was a gradient of Peak G values from the apex to the base of LV (Figure 5). The time to peak G did not differ significantly among most segments. However, the median time to peak G in the apical segments of the posterior, lateral, and septal walls was significantly longer compared to the basal and middle segments of the anterio-septal and septal walls (Appendix A). As for the peak S, no significant correlation was found between peak G and the time to peak G. Compared with the global median time to peak G, statistically longer times were observed in the lateral wall segments at the middle and apical levels and the septum at the apical level. The anterio-septal wall segment at the basal level statistically reached peak G earlier than the global value (Appendix A). For most segments of the LV, a significant positive correlation was found between the P peak value and the G peak value; the time to peak P was positively correlated with the time to peak G (Appendix A).

### 3.2. Circumferential Strain

Figure 6, Figure 7 and Figure 8 illustrate the distributions of peaks S, G, and P, as well as the time to peaks S, G, and P, relative to the location of LV segments for circumferential strain.

The absolute value of peak S of circumferential strain (Figure 6) was statistically significantly greater in the anterior septal and septal walls than in the other LV walls (Appendix A). Significantly greater absolute values of the circumferential strain were found compared to the global median in all segments of the anterio-septal wall, septum, and apical segment of the inferior wall (Appendix A). Significantly smaller absolute values compared to the global median were observed in the segments of the lateral wall, basal and medial segments of the posterior wall, and middle and apical segments of the anterior wall (Appendix A).

The time to peak S (Figure 6) for most segments did not differ statistically among themselves (Appendix A), with the exception of the basal segments of the anterior and lateral walls, where the median time to peak was minimal and significantly different from the basal segment of the inferior wall, the middle segment of the posterior wall, and the apical segments of the lateral, posterior, and inferior walls (Appendix A). For circumferential strain, no significant correlation was found between peak S and the time to peak S (Appendix A).

In more than half of the subjects in the study group, the peak P value in the circumferential direction was equal to 0 in the segments of the anterio-septal wall and septum. The maximum median values were observed at the basal, middle, and apical levels in the segments of the anterior wall (Figure 7, Appendix A). Significantly higher peak P values compared to the global median were observed in the anterior, lateral, and posterior segments, while a significantly lower peak P value was observed in the middle segment of the septum (Appendix A).

The time to peak P was significantly longer than the global median in the anterior wall segments, the basal segments of the lateral wall and septum, and the middle and apical segments of the anterio-septal wall, while the time to peak was significantly shorter than the global median in the posterior and inferior wall segments (Appendix A).

In the circumferential direction, we did not find a significant correlation between peak P value and the time to peak P. A significant positive correlation was found between peak P and S values in all segments except the basal and middle septal segments (see Appendix A). The time to peak S was positively correlated with the time to peak P in the basal and apical segments of the inferior wall and in the middle and apical segments of the posterior wall (Appendix A).

As for longitudinal strain, a significant positive correlation was found between peak S and peak G, time to peak S and time to peak G for all LV segments (Appendix A).

Similarly to the peak S in the circumferential direction, the absolute value of the peak G was statistically significantly greater for the anterio-septal and septal wall compared to other LV walls (Figure 8). However, the time to peak G in the segments did not differ significantly, with the exception of the apical septum segment, where the time to peak was maximal compared to other segments (Appendix A). No significant correlations were found between peak G and the time to it (Appendix A).

Longitudinal strain in the cardiac cycle was significantly higher compared to circumferential strain in most LV segments, except in the basal segments of the lateral, inferior wall, and basal and mid-septal segments, where no significant differences were found. The only basal segment of the anterio-septal wall had significantly lower longitudinal strain compared to circumferential strain. The time to reach the longitudinal strain peaks was significantly longer than the time to reach the circumferential strain peaks, except for the anterio-septal, inferior, and septal segments, where the time was not significantly different (Appendix A).

### 3.3. Radial Strain

Figure 9 shows the distributions of peak G and time to peak G relative to LV wall location for radial strain. The Supplement provides percentile tables of the parameters under consideration.

Peak G demonstrated statistically significantly greater values of radial strain in the septal wall compared to other LV walls (Figure 9). Compared to the global median, significantly greater values were observed in all segments of the septum, basal and middle segments of the inferior wall, and apical segment of the anterio-septal wall. The time to peak G in the radial direction did not differ significantly between segments and did not differ significantly from the global median time to peak G.

The main results are summarized schematically in Figure 10.

## 4. Discussion

The complex geometry and structure of the human LV are closely related to the spatio-temporal sequence of its electrical activation and regional contraction, which is necessary for an effective pumping function. It has been shown that the spatio-temporal dynamic change in the LV configuration during the heart cycle plays a significant role in regulating the mechanical and pumping function of the LV [12]. In an adult population, a significant change in the regional function of the LV has been demonstrated in heart disease, and assessing the regional features of LV deformation can be important for the diagnosis, prognosis, and treatment of various heart disease [13,14].

Research on the deformation of heart chambers using the STE method in premature infants during the early neonatal period is scarce. The sources we found in global scientific publications only provide values for global strain, most often GLS. In addition, only one study evaluated the more accurate (from three apical views) global longitudinal strain of the LV [6], while other studies used only a four-chamber view [7,15,16,17,18]. There are also few studies that obtained parameters in the first 72 h of life in premature infants weighing less than 1500 g [6,15,17].

We decided to study segmental strain in more detail and establish a range of reference values, including because our research group had previously recorded cases of wall akinesia/hypokinesia (visual assessment) in premature infants with preserved global systolic LV function at birth, which may indicate the development of myocardial infarction. This pathology is rare in infants. The causes of myocardial infarction in infants and young children are diverse. The literature reports cases of myocardial infarction in infants with congenital heart and coronary artery abnormalities, severe perinatal hypoxia, respiratory distress syndrome, broncho-pulmonary dysplasia, and primary pulmonary hypertension. Severe hypoxia can lead to small-scale intramural infarctions, with disseminated intravascular coagulation playing a significant role in their development. The leading cause of myocardial infarction in the neonatal period may also be paradoxical umbilical vein embolism [19]. Undoubtedly, a specialized study of this group of infants is needed, which will be a further step in our research. Currently, having even basic knowledge of the reference values for segmental LV strain simplifies the diagnosis of segmental anomalies during screening echocardiography.

For the first time, we conducted a complete study of longitudinal, circumferential, and radial LV strain using the STE in extremely premature infants without signs of hemodynamic disorders in the first 72 h of life.

In the population we examined, the GLS values, which we estimated using three apical views, were −16.4 (−14.5; −18.4), slightly lower compared to the only study that also used three apical views to estimate GLS (−18.4 (−14.6; −22.2)). However, we can see that the range of values we observed intercepts with the range of data reported in [6] having a larger sample size (239 infants) than that (65 infants) in our study.

In addition to the assessment of GLS, the following longitudinal, circumferential, and radial strain characteristics were obtained: peak S, peak G, peak P, and the time to reach them in 18 segments of the LV (according to the scheme in Figure 2).

In the cohort we considered, we did not identify significant correlations of the strain characteristics under consideration with birth weight and gestational age. A similar result was obtained in the study [6], where no statistically significant linear correlation dependence was found between birth weight, gestational age and LV GLS, free wall longitudinal strain (r = 0.34 and 0.44, *p* > 0.1). However, a statistically significant correlation dependence was reported for the right ventricle GLS and septum LS.

We also checked whether there were any differences in the deformation parameters between infants with an open and closed arterial duct in the group under consideration. No statistically significant differences were found in the deformation characteristics, which can be explained by the absence of volume overload in the pulmonary circulation during the first few days of life due to increased pulmonary vascular resistance. In the presence of patent ductus arteriosus, at older ages, when the ductus becomes hemodynamically significant, LV overload occurs, which can alter deformation parameters (from improvement to deterioration). This hypothesis requires further study.

Longitudinal shortening of myocardial fibers contributed more to LV contraction than circumferential shortening. This was because the value of longitudinal strain peaks and the time to reach them were significantly greater than those of circumferential strain for most LV segments.

The apical segments exhibited significantly greater strain in the longitudinal direction than in the circumferential direction. The basal segments of the inferior and septal walls, as well as the middle septum, underwent uniform deformation in the longitudinal and circumferential directions. The only exception was the basal segment of the anterio-septal wall, which underwent greater circumferential strain.

Segmental longitudinal strain was heterogeneous, with significant quantitative differences in the contribution of different LV segments to global LV longitudinal deformation. The longitudinal strain of the apical segments exceeded the global value, whereas the strain of the basal and middle septal segments was significantly lower.

We revealed a gradient of longitudinal strain from the apex to the base of LV. Thus, statistically significant, higher absolute values of peak S and G were observed in most apical segments compared to middle and basal segments. Similar results were described in the work of Levy PT et al. [6], in which the longitudinal strain of the interventricular septum, LV free wall, and right ventricular free wall was studied. They showed that, in children, there was a negative gradient of longitudinal strain from the apex to the base of the LV; that is, LV strain was most pronounced in the apical region compared to the base. This gradient presumably arises from the convergence of the right-sided and left-sided spirals of myocardial fibers in the sub-endocardium and sub-epicardium, respectively, toward the apex of the heart. This formation is known as a “double helix loop vortex.” Additionally, the apex has a thinner wall and greater curvature than the middle and base regions of LV, resulting in higher stress in the apical region compared to other LV regions [6].

The positive peak P of longitudinal strain, which characterizes regional pre-stretching at the onset of systole, differed significantly for segments of the lateral and posterior walls from the global peak P characteristic and the median zero value observed in the basal segments of the anterio-septal and septal walls of the left ventricle (LV). Furthermore, a significant positive correlation was found between peak P value and time to reach it. In other words, the lateral and posterior wall segments were pre-stretched to a greater extent and for a longer period compared to other LV segments. Accordingly, shortening in these regions began later. Early systolic prolongation in the lateral and inferior walls may be associated with their relative weakness compared to the stronger adjacent walls (the septum, anterio-septal walls). At the beginning of systole, these stronger walls contract and pull the weaker walls, causing them to stretch.

Additionally, a significant positive correlation was found for most segments between the time to peak P and the time to peak S and G. In other words, the lateral and posterior wall segments underwent longer pre-stretching and began to shorten later than the other regions. These segments reached their peak longitudinal strain later. The strain of lateral and posterior wall segments at each level (MV, PM, and AP) was smaller than in other segments at the same level. In other words, segments that were pre-stretched earlier tended to reach peak strain earlier and at a greater magnitude than segments that reached peak strain later. This statistically significant relationship was established in full-term neonates using functional geometry methods [20].

Compared to other segments of the LV and the global strain characteristic, the anterio-septal and septal segments exhibited greater strain value in the circumferential direction. However, the time to peak strain in these segments did not differ significantly. The median value of early systolic elongation (peak P) in the anterio-septal and septal segments was zero. In other words, in the circumferential direction and in the absence of pre-stretch, greater strain peaks were reached in the same amount of time as in other segments of the LV. This is consistent with the fact that, during LV activation, the septum is activated more uniformly over time due to the presence of the conduction system [12].

In the radial direction, the anterio-septal, septal, and inferior wall segments were identified as the areas with the greatest LV wall thickening. In other words, the septal and anterio-septal wall segments thicken and develop greater tension in the circumferential direction during systole compared to other LV segments.

This study examined the characteristics of segmental, circumferential, and radial LV strain in extremely preterm infants without haemo-dynamic compromise during the first 72 h of life. We hypothesize that strain values outside the reference range may indicate regional myocardial pathology. Further investigation is required to confirm this hypothesis.

## 5. Conclusions

This is the first study to examine the patterns and establish the reference values of longitudinal, circular, and radial LV strain in extremely premature infants without hemodynamic disorders during the first 72 h of life using STE.

Further research should focus on standardizing values and studying the long-term prognosis in children with altered strain parameters.

### Limitation

Performing STE in this group of children is challenging due to their small body surface area, motor activity, lack of conscious interaction, and high heart rate. Two children were excluded from the group because we were unable to evaluate their STE parameters.

Image scanning and parameter calculations are performed on a GE Vivid iq ultrasound system. Comparisons with results obtained on other equipment and software have not been performed.

All images were acquired by an individual operator; inter-operator variability was not assessed.

## Figures and Tables

**Figure 1 pediatrrep-17-00126-f001:**
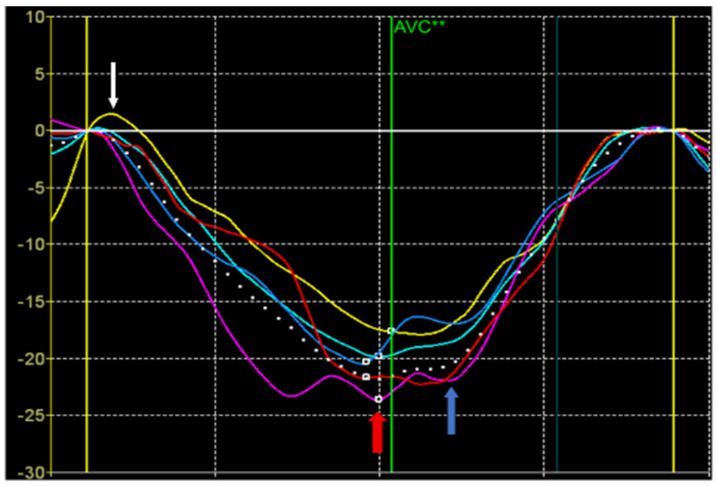
The segmental strain traces derived from the LV apical four-chamber view. The colored lines show the time-dependent strain change during the cardiac cycle in each segment. Peak P is indicated by the white arrows, Peak S—red arrows, and Peak G—blue arrows. AVC**—aortic valve closure. *X*-axis—time (ms); *Y*-axis—strain (%).

**Figure 2 pediatrrep-17-00126-f002:**
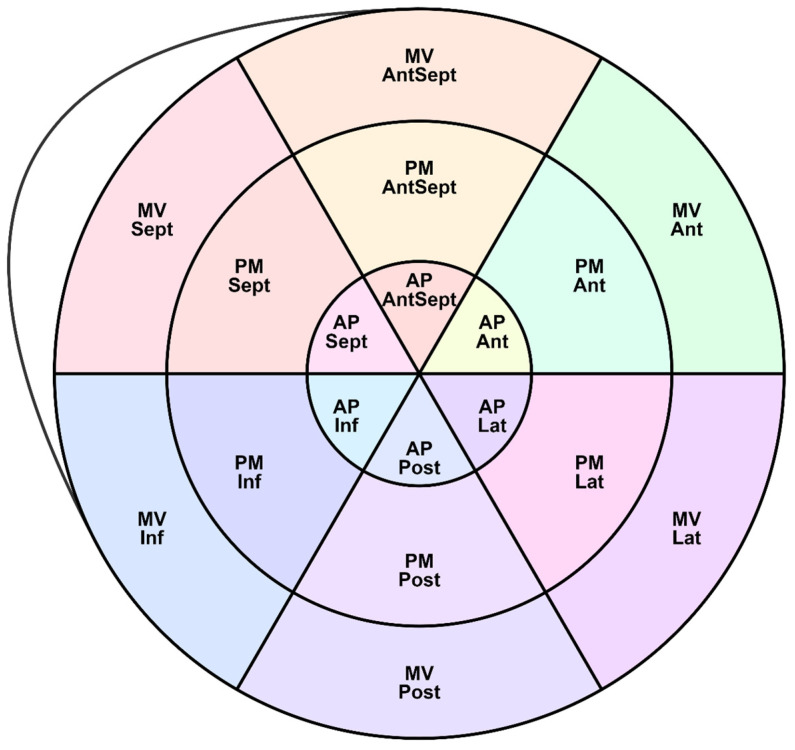
18-segment scheme of the LV. MV—mitral valve level, PM—papillary muscles level, AP—apical level, Ant—anterior wall, Lat—lateral wall, Post—posterior wall, Int—inferior wall, Sept—septal wall, AntSept—anterio-septal wall.

**Figure 3 pediatrrep-17-00126-f003:**
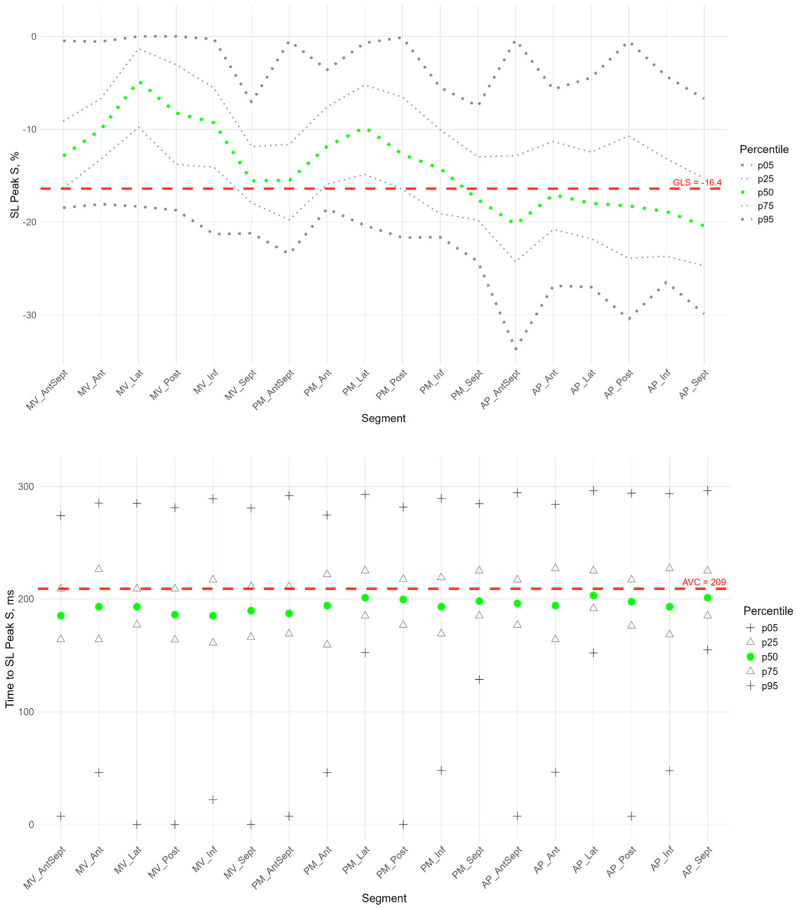
Distribution of SL Peak S (**top**) and Time to SL Peak S (**bottom**) relative to LV segments.

**Figure 4 pediatrrep-17-00126-f004:**
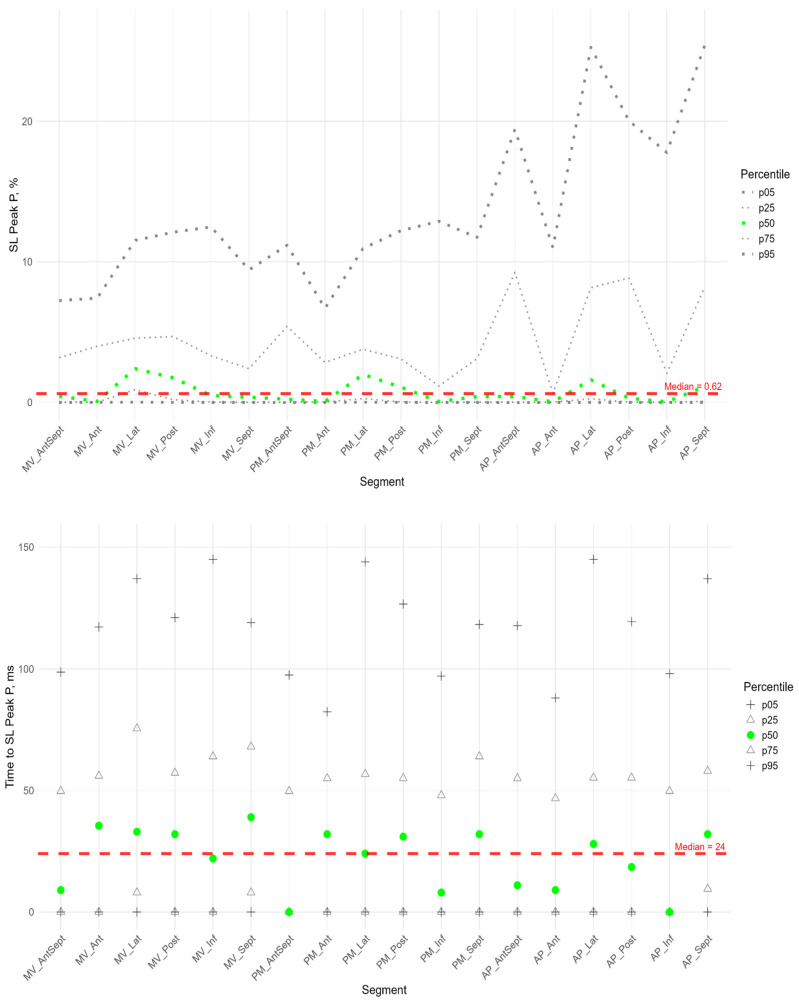
Distribution of SL Peak P (**top**) and Time to SL Peak P (**bottom**) relative to LV segments.

**Figure 5 pediatrrep-17-00126-f005:**
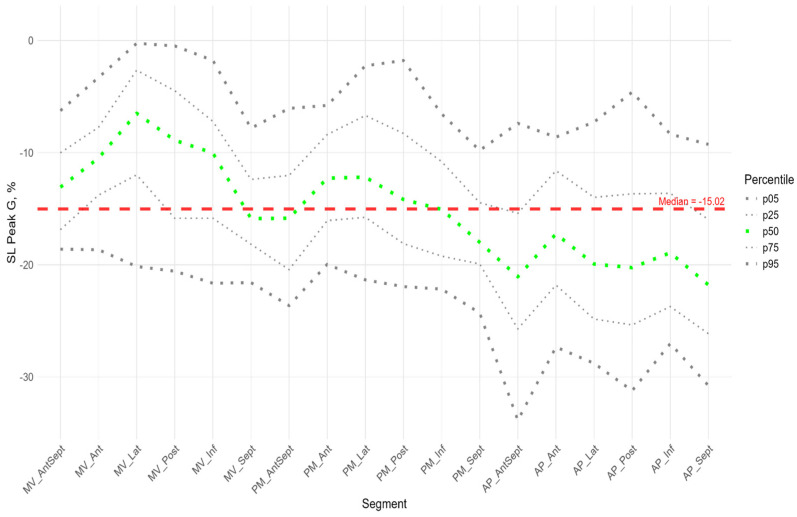
Distribution of SL Peak G (**top**) and Time to SL Peak G (**bottom**) relative to LV segments.

**Figure 6 pediatrrep-17-00126-f006:**
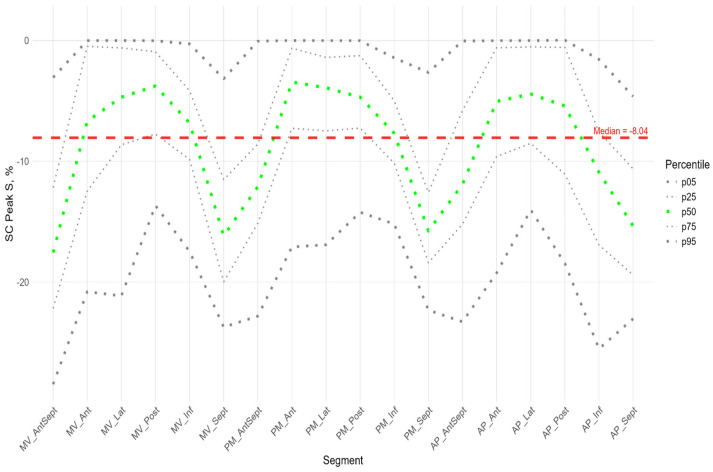
Distribution of SC Peak S (**top**) and Time to SC Peak S (**bottom**) relative to LV segments.

**Figure 7 pediatrrep-17-00126-f007:**
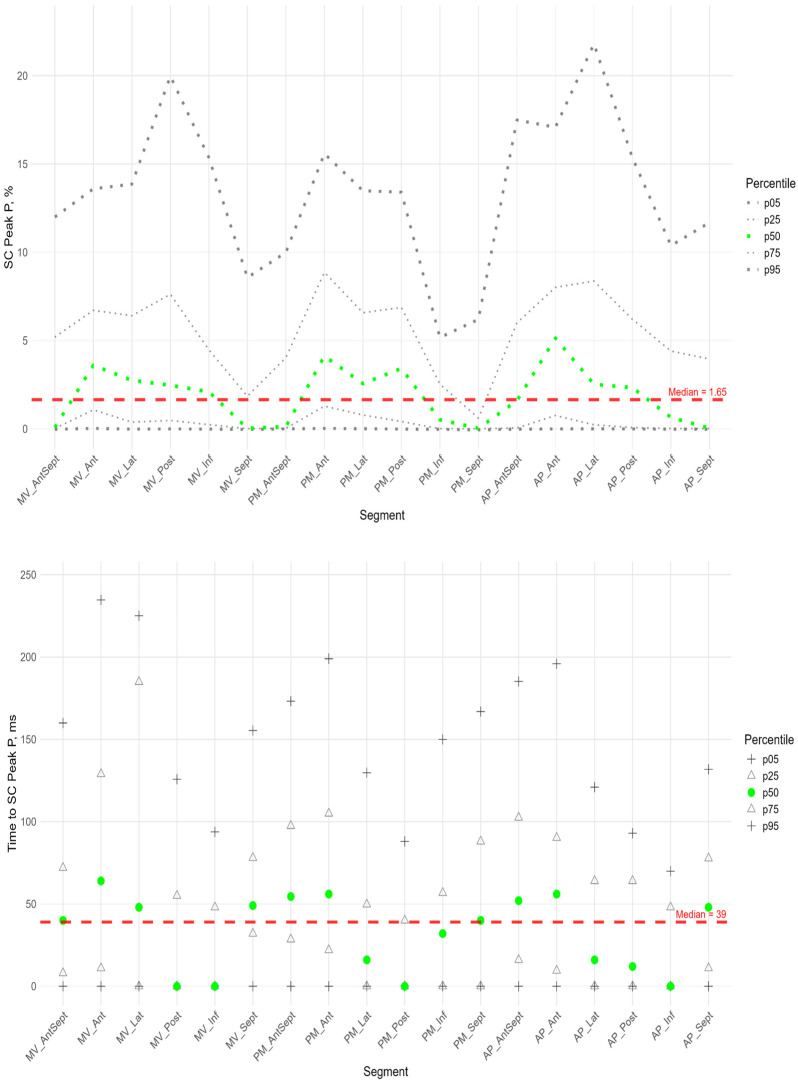
Distribution of SC Peak P (**top**) and Time to SC Peak P (**bottom**) relative to LV segments.

**Figure 8 pediatrrep-17-00126-f008:**
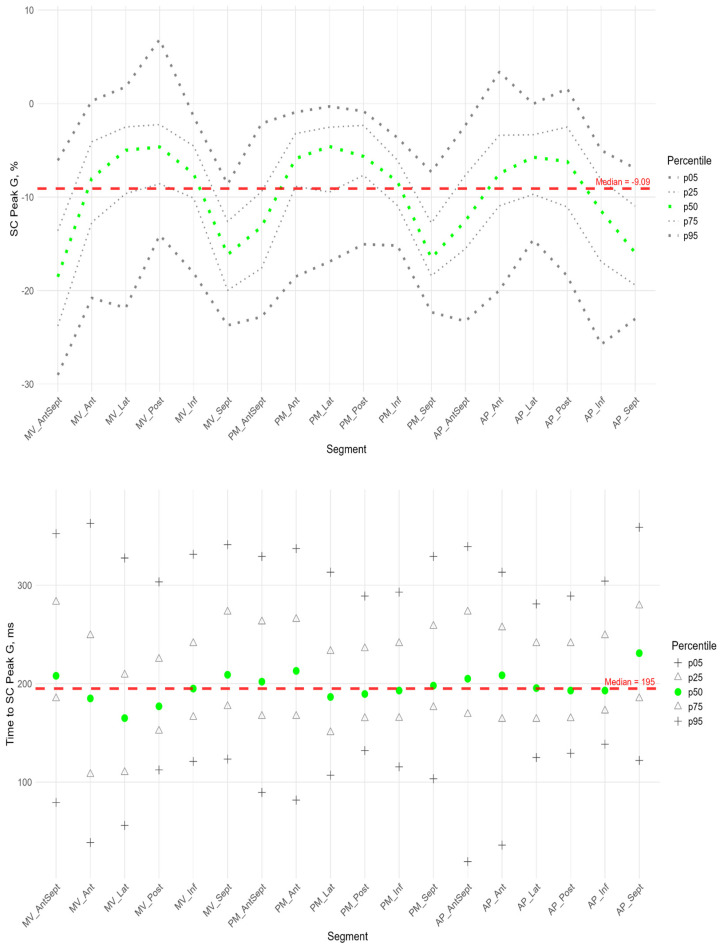
Distribution of SC Peak G (**top**) and Time to SC Peak G (**bottom**) relative to LV segments.

**Figure 9 pediatrrep-17-00126-f009:**
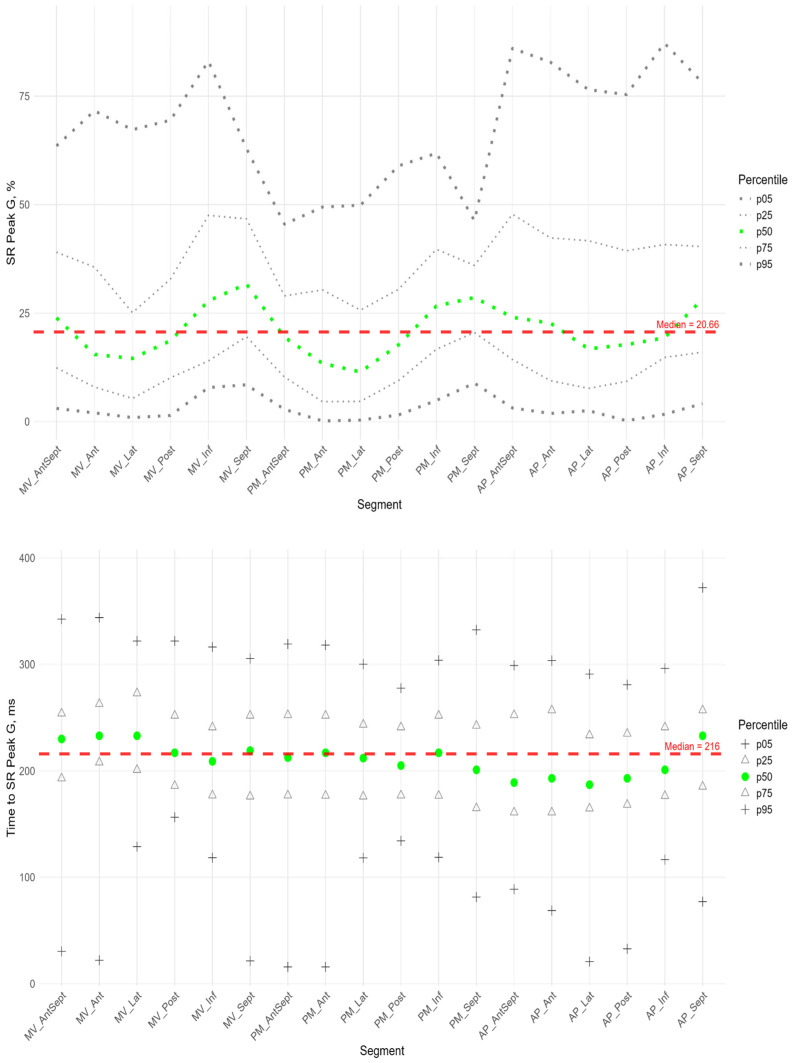
Distribution of SR Peak G (**top**) and Time to SR Peak G (**bottom**) relative to LV segments.

**Figure 10 pediatrrep-17-00126-f010:**
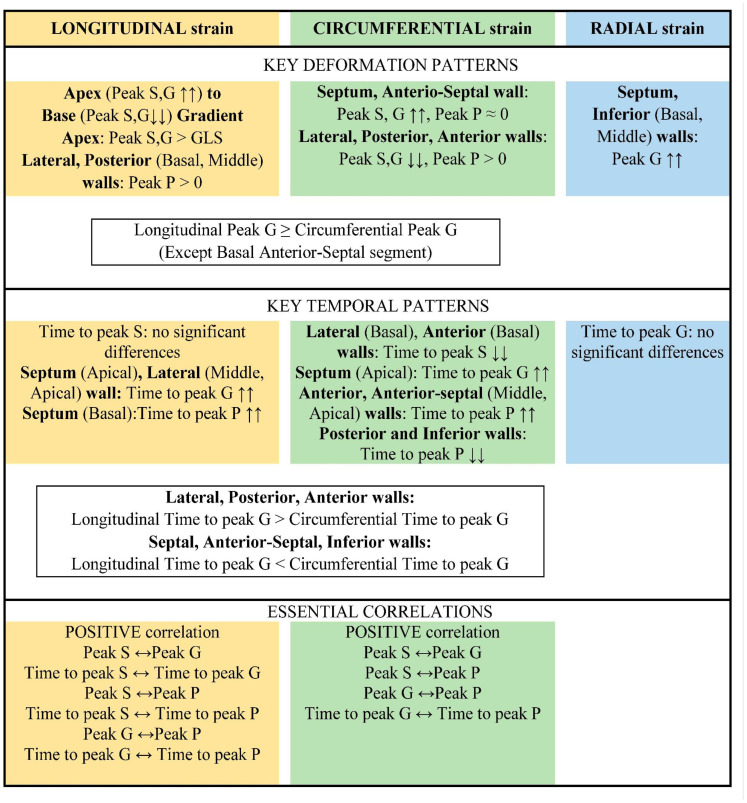
Key findings in the results. ↑↑—the highest absolute value among the LV segments, ↓↓—the lowest absolute value among the LV segments, GLS—global longitudinal strain, ↔—significant correlation.

**Table 1 pediatrrep-17-00126-t001:** Description of a group of extremely premature newborns without signs of hemodynamic disturbances (*n* = 65).

Parameters	Preterm (*n* = 65) Me (25; 75%)
Gestation age, weeks	29.7 (27.4; 31.1)
Weight, g	1210 (980; 1420)
Height, cm	37 (35; 40)
Age, hours	41(21; 56)
Mean blood pressure, mmHg	47 (43; 56)
Heart rate, bpm	155 (147; 166)
Saturation (SpO_2_), %	96 (94; 98)
Patent ductus arteriosus, *n* (%)	46 (71%)
Intraventricular haemorrhage, *n* (%)	14 (22%)
**Respiratory support**	
Respiratory support, *n* (%)	62 (95.4%)
Mechanical ventilation, *n* (%)	23 (35.4%)
Hight frequency ventilation, *n* (%)	22 (33.9%)
CPAP, *n* (%)	16 (24.6%)
HFNC, *n* (%)	1 (1.5%)
**Indications for delivery**	
fetal growth restriction	7 (10.8%)
Preeclampsia	13 (20.0%)
Bleeding	11(16.9%)
Chorioamnionitis	8 (12.3%)
Premature rupture of membranes	14 (21.5%)
Premature birth without associated diagnosis	12 (18.5%)
**Echocardiography**	
End diastolic diameter LV, mm	13 (11; 15)
End systolic diameter LV, mm	8.5 (7; 10)
Ejection fraction LV (Teicholz), %	68 (63; 72)
Thickness of the interventricular septum, mm	3.7 (3; 4)
Thickness of the posterior wall LV, mm	2 (1; 2)
Cardiac output, min∙mL/kg	252 (201; 303)
Cardiac index by VTI	2.8 (2.1; 3.3)
**Global strain parameters**	
LV GLS, %	−16.4 (−14.5; −18.4)
AVC, ms	209 (193; 225)
Global median Time to SL peakS, ms	195 (169; 219)
Global median SL peakG, %	−15 (−20; −10)
Global median Time to SL peakG, ms	225 (185; 257)
Global median SL peakP, %	0.6 (0; 4)
Global median Time to SL peakP, ms	24 (0; 56)
Global median SC peakG, %	−9 (−15; −5)
Global median Time to SC peakG, ms	195 (161; 252)
Global median SC peakS, %	−8 (−15; −3)
Global median Time to SC peakS, ms	176 (99; 198)
Global median SC peakP, %	2 (0; 6)
Global median Time to SC peakP, ms	39 (0; 77)
Global median SR peakG, %	21 (11; 38)
Global median Time to SR peakG, ms	216 (176; 252)

Abbreviations: CPAP, constant positive airway pressure; HFNC, high-flow nasal cannula; LV, left ventricular; VTI, velocity-time integral; LV GLS, left ventricular global longitudinal strain; AVC, aortic valve closure; SL, longitudinal strain; SC, circumferential strain; SR, radial strain.

## Data Availability

The raw data supporting the conclusions of this article will be made available by the authors on request. The data are not publicly available due to privacy.

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
