# Peer review of "Patterns of Segmental Strain of the Left Ventricle in Extremely Premature Infants"

_pediatrrep, 2025, doi:10.3390/pediatric17060126_

Round 1
Reviewer 1 Report
Comments and Suggestions for Authors
The manuscript is a necessary addition to the scientific literature that deals with extremely preterm infants, since it provides sensitive data and reference values regarding myocardial strain. That being said, I merely wish to point out some minor issues I found in my critical read of this manuscript:
- The title is too long – the authors wanted to be very precise regarding the topic, but in doing so, they came up with a title that is difficult to read and can put off readers
- In the Results section, the authors claim that all infants had postnatal ages less than 72 hours, but a median postnatal age should also be given
- In the Limitations subsection, the authors mention two infants being excluded from the study – this should be mentioned in the Material and Methods section
- Reference 11 is incorrectly cited
- There are a few issues with the English language throughout the manuscript, here are some suggestions: ”large range” instead of ”range”, on line 23, ”impairment” instead of ”disturbancies” on line 177, ”correlations” instead of ”dependencies” on line 186, ”condition” instead of ”pathology” on line 374.
Author Response
Comments 1: [The title is too long – the authors wanted to be very precise regarding the topic, but in doing so, they came up with a title that is difficult to read and can put off readers.]
Response 1: Thank you for pointing this out. We changed the title of the article to [“Patterns of segmental strain of the left ventricle in extremely premature infants”]
Comments 2: [In the Results section, the authors claim that all infants had postnatal ages less than 72 hours, but a median postnatal age should also be given.]
Response 2: Agree. We thank the reviewer for this suggestion. The age of the children was 41 hours (21; 56), these data are included in Table 1.
Comments 3 : [In the Limitations subsection, the authors mention two infants being excluded from the study – this should be mentioned in the Material and Methods section]
Response 3: Thank you for your comment. We have added this information to line # 94-96.
Comments 4: [Reference 11 is incorrectly cited]
Response 4: Thank you. We've provided a more recent reference. Now this reference is number 12. We have added this information to line # 615-617.
Response to Comments on the Quality of English Language
Point 1: There are a few issues with the English language throughout the manuscript, here are some suggestions: ”large range” instead of ”range”, on line 23, ”impairment” instead of ”disturbancies” on line 177, ”correlations” instead of ”dependencies” on line 186, ”condition” instead of ”pathology” on line 374.
Response : Thanks for the comments, everything has been corrected.
Reviewer 2 Report
Comments and Suggestions for Authors
1.The introduction mentions that segmental strain can detect pathology even with preserved global function, and the discussion briefly references observed wall motion abnormalities. This critical point should be emphasized earlier and more concretely.
2.The parameters Peak S, Peak G, and Peak P are central to the analysis, but their physiological significance is not fully explained, particularly Peak P (prestretch).
3.The discussion of the longitudinal prestretch (Peak P) in the lateral and posterior walls is intriguing. The manuscript could more deeply explore the potential clinical or physiological implications of this finding.
Author Response
Comments 1: [The introduction mentions that segmental strain can detect pathology even with preserved global function, and the discussion briefly references observed wall motion abnormalities. This critical point should be emphasized earlier and more concretely.]
Response 1: [Type your response here and mark your revisions in red] We thank the reviewer for this suggestion. We included additional information in the edited manuscript (lines [41-49]).
[The biplane Simpson method for assessing global LV function results in high inter- and intra-operator variability and low reproducibility. A more accurate method for assessing LV systolic function is the global longitudinal strain, which allows for the identification of minor changes at earlier stages [1]. It is suggested that segmental strain testing may help in detecting focal lesions of the LV myocardium, such as those associated with neonatal coronary pathology (coronary artery origin anomalies, thrombosis, fibroelastosis, etc.). In the presence of segmental pathology, the function of the affected segment may be compensated for by the increased function of healthy adjacent segments, and global systolic dysfunction indices may be within normal limits [2]]
1.Mertens L, Singh G, Armenian S, Chen MH, Dorfman AL, Garg R, Husain N, Joshi V, Leger KJ, Lipshultz SE, Lopez-Mattei J, Narayan HK, Parthiban A, Pignatelli RH, Toro-Salazar O, Wasserman M, Wheatley J. Multimodality Imaging for Cardiac Surveillance of Cancer Treatment in Children: Recommendations From the American Society of Echocardiography. J Am Soc Echocardiogr. 2023 Dec;36(12):1227-1253. doi: 10.1016/j.echo.2023.09.009. PMID: 38043984.
2.Tyurina, L.G.; Khamidova, L.T.; Ryubalko, N.V.; Koltashova, S.A.; Kislukhina, E.V.; Gazaryan, G.A. Potential of speckle tracking echocardiography with the assessment of left ventricle myocardial work in predicting coronary artery disease in non-ST-segment elevation acute coronary syndrome. Medical Alphabet 2024, (14), pp. 33–42. (In Russ.) https://doi.org/10.33667/2078-5631-2024-14-33-42
Comments 2: [The parameters Peak S, Peak G, and Peak P are central to the analysis, but their physiological significance is not fully explained, particularly Peak P (prestretch)]
Response 2: We have added a discussion on this aspect in the [Material and Methods] section ( lines [126-147]) to provide more clarity.
[Peak S of longitudinal or circumferential strain was defined as the maximum negative value during the systole. Peak S is defined on the deformation curve in the systole before the closure of the aortic valve. It reflects the maximum shortening of the segment in systole and is used to assess the contribution of the segment to the pumping function of the heart.
Peak G was defined as the maximum absolute value of longitudinal, circumferential, and radial deformation throughout the cardiac cycle. Peak G can occur either before or after aortic valve closure. If the time position of the peak G is before aortic valve closure, the values ​​of the peak S and peak G coincide. A peak G positioned after aortic valve closure suggests a pattern of postsystolic shortening. Postsystolic shortening may be classified as physiological if its amplitude and duration are small. It is believed that the presence of this pattern with large amplitude and duration indicates an enhancement of myocardial stiffness at the beginning of diastole, including the period of isovolumetric relaxation.
Peak P of longitudinal or circumferential strain was defined as the positive systolic strain peak. If there is no positive peak, it is set equal to 0. Peak P occurs at the beginning of systole, including the phase of isovolumetric contraction with closed valves. Peak P indicates early systolic lengthening of the segment before the onset of its shortening during systole. Normally, it may be observed in normal segments having a small amplitude and duration. In pathology, the duration and amplitude of the P peak may increase and occupy the entire systole indicating a significant damage of the myocardial segment [11].]
Comments 3 : [The discussion of the longitudinal prestretch (Peak P) in the lateral and posterior walls is intriguing. The manuscript could more deeply explore the potential clinical or physiological implications of this finding.]
Response 3:Thank you for your comment. We have added this information (lines [511-514]).
[Early systolic prolongation in the lateral and inferior walls may be associated with their relative weakness compared to the stronger adjacent walls (the septum, anterio-septal walls). At the beginning of systole, these stronger walls contract and pull the weaker walls, causing them to stretch.]
Reviewer 3 Report
Comments and Suggestions for Authors
This manuscript offers a technically advanced and clinically meaningful assessment of left ventricular (LV) deformation mechanics in extremely premature neonates using speckle-tracking echocardiography (STE). The topic is both timely and significant, as myocardial mechanics in this fragile population remain insufficiently characterized. The authors provide valuable reference percentile data and delineate notable strain gradients.
That said, the paper would benefit from clearer presentation, sharper focus, and stronger contextual integration with existing neonatal imaging literature. It is potentially publishable pending minor revisions to refine its scientific framing, figures, and interpretative depth.
-
The prospective study design is appropriate, but inclusion and exclusion criteria should be described in greater detail (e.g., specify how “no hemodynamic impairment” was defined—by echocardiography, clinical findings, or hemodynamic indices).
-
Clarify whether angle correction, manual ROI adjustment, and endocardial tracking quality scoring were implemented.
-
The reported intra-observer reliability (ICC = 0.96) is excellent, though including inter-observer reproducibility would further strengthen methodological rigor.
-
Percentile-based data presentation is suitable, but the rationale for selecting percentile distributions should be explicitly stated.
-
Condense Supplementary Tables S1–S6 into concise summary tables within the main text and avoid repetitive “see Supplement…” references.
-
The Results section is overly descriptive; summarize core quantitative findings (e.g., GLS range, magnitude of base-to-apex gradient) more succinctly.
-
In the Discussion, compare the observed mean GLS (≈ –16.4%) with prior neonatal cohorts more systematically.
-
Acknowledge vendor and software dependency as a methodological limitation.
-
Offer a more balanced explanation for the absence of correlation with birth weight or gestational age—whether due to small sample size, physiological homogeneity, or limited parameter variability.
Author Response
Comments 1: [The prospective study design is appropriate, but inclusion and exclusion criteria should be described in greater detail (e.g., specify how “no hemodynamic impairment” was defined—by echocardiography, clinical findings, or hemodynamic indices).]
Response 1: [Type your response here and mark your revisions in red] We thank the reviewer for this suggestion. We now have additional information in the edited manuscript (lines [90-93]).
Comments 2: [Clarify whether angle correction, manual ROI adjustment, and endocardial tracking quality scoring were implemented.]
Response 2: We have added a discussion on this aspect in the [Material and Methods] section ( lines [113-120]) to provide more clarity.
[Images were acquired with the direction angle most parallel to the region of interest (ventricular walls), in accordance with speckle tracking recommendations, owing to the improved spatial resolution along the ultrasound beam. The contouring zone was superimposed using the reference point placement method. To achieve more accurate results, the contouring area was carefully checked and manual corrections were made if necessary. Tracking quality was assessed automatically by the software. If inadequate tracking was detected, additional manual corrections were performed. Endocardial tracking quality scoring was not performed.]
Comments 3 : [The reported intra-observer reliability (ICC = 0.96) is excellent, though including inter-observer reproducibility would further strengthen methodological rigor.]
Response 3: Thank you for your comment. All images were acquired by a single operator, so interoperator variability was not assessed. The following sentence has been added to the "Limitations" section, lines [557-558].
Comments 4 : [Percentile-based data presentation is suitable, but the rationale for selecting percentile distributions should be explicitly stated.]
Response 4:Thank you for your comment. We have added a discussion on this aspect in the [Material and Methods] section (lines [189-192]) to provide more clarity.
Percentiles were chosen to represent the data because this approach provides a robust and intuitive understanding of its distribution. Percentiles do not make any assumptions about the distribution of the data and simply describe it as it is. Presenting key percentiles (e.g., P5, P25, P50, P75, P90, P95) clearly shows where the data is concentrated and what the tails of the distribution look like. Percentiles are ideal for understanding what is "acceptable" for the majority of a population. Percentiles are easier for medical audiences to understand than concepts like variance or standard deviation. For example, the presentation of anthropometric data (weight, height, head circumference) in the form of percentile tables or curves is chosen because this method allows individual data to be assessed in the context of the entire population of children of the same gestational age.
Comments 5 : [The reported intra-observer reliability (ICC = 0.96) is excellent, though including inter-observer reproducibility would further strengthen methodological rigor.]
Comments 6 : [The Results section is overly descriptive; summarize core quantitative findings (e.g., GLS range, magnitude of base-to-apex gradient) more succinctly.]
Response 5,6: Thank you for your comments.
The data in the supplement tables are extensive and cannot be included in the main text of the article; a summary of each table is given in the text of the results with the corresponding reference.
We would like to leave most of the existing references for the reader, given the volume and complexity of the research materials.
We've removed unnecessary references and more precisely formulated the summary for each table. We've also added a summary diagram of the results (Fig.10).
Comments 7 : [In the Discussion, compare the observed mean GLS (≈ –16.4%) with prior neonatal cohorts more systematically.]
Response 7: In the population we examined, the GLS values, which we estimated using three apical views, were -16.4 (-14.5; -18.4), slightly lower compared to the only study that also used three apical views to estimate GLS (-18.4 (-14.6; -22.2)). However, we can see that the range of values we observed intercepts with the range of data reported in [6] having a larger sample size (239 infants) than that (65 infants) in our study. The sample size in this method was larger - 239 newborns, which is probably why we obtained different values.
We have added a discussion on this aspect in the [Discussion] section (lines [456-460]).
Comments 8: [Acknowledge vendor and software dependency as a methodological limitation.]
Response 8: Thank you for your comments. The following wording has been added to the "Limitations" section, lines [554-556]: [Image scanning and parameter calculations are performed on a GE Vivid iq ultrasound system. Comparisons with results obtained on other equipment and software have not been studied.]
Comments 9: [Offer a more balanced explanation for the absence of correlation with birth weight or gestational age—whether due to small sample size, physiological homogeneity, or limited parameter variability.]
Response 9: Thank you for your comments.
In a study on a child population (n=103) from 1 to 18 years old, where the relationship between weight, age and speckle tracking characteristics was investigated, such a relationship was not statistically confirmed [1]. We also provide a comparison with a study on a similar population in the text of the article, which also found no significant correlation between gestational age and birth weight [2]. We do not deny that this may be due to the small sample size and/or physiological homogeneity, which requires further research.
1.Koopman LP, Rebel B, Gnanam D, Menting ME, Helbing WA, Boersma E. (2019) Reference values for two-dimensional myocardial strain echocardiography of the left ventricle in healthy children. Cardiology in the Young29: 325–337. doi: 10.1017/S1047951118002378
- Levy, P.T.; El-Khuffash, A.; Patel, M.D.; Breatnach, C.R.; James, A.T.; Sanchez, A.A.; Abuchabe, C.; Rogal, S.R.; Holland, M.R.; McNamara, P.J.; Jain, A.; Franklin, O.; Mertens, L.; Hamvas, A.; Singh, G.K. Maturational patterns of systolic ventricular deformation mechanics by two-dimensional speckle-tracking echocardiography in preterm infants over the first year of age. J. Am. Soc. Echocardiogr. 2017, 30(7), pp. 685–698.e1. https://doi.org/10.1016/j.echo.2017.03.003
We have added a discussion on this aspect in the [Discussion] section (lines [465-469]).
Reviewer 4 Report
Comments and Suggestions for Authors
The study offers the first comprehensive dataset of segmental myocardial strain values in extremely premature infants using STE. The topic is clinically relevant, methodologically rigorous, and will be of great interest to pediatric cardiology readers.
Overall, the study is well-designed and executed. However, several aspects could be improved to enhance clarity, reproducibility, and interpretability. My detailed comments are as follows:
- Please include more detail about the image acquisition process, especially how adequate frame rates and tracking quality were ensured given the high heart rates and small body size of preterm infants. Clarify how poor-quality or incomplete segments were handled during analysis. Mention if inter-observer variability was evaluated or, if not, acknowledge it as a limitation.
- Specify any sedation policy or handling technique during imaging to minimize motion artifacts. Provide a brief justification for sample size (e.g., power estimate or precedent in similar studies).
- The results are comprehensive but somewhat dense. Consider summarizing key quantitative findings (e.g., representative median strain values or base-to-apex gradients) in the main text rather than referring extensively to supplementary material.
- The discussion is thorough and well-informed. However, you could further emphasize the clinical applicability of your reference values—how might clinicians use these data to identify early myocardial dysfunction?
- Please discuss potential effects of physiological variability (e.g., PDA status, respiratory support type) on strain values, even if not statistically significant.
Author Response
Comments 1: [Please include more detail about the image acquisition process, especially how adequate frame rates and tracking quality were ensured given the high heart rates and small body size of preterm infants. Clarify how poor-quality or incomplete segments were handled during analysis. Mention if inter-observer variability was evaluated or, if not, acknowledge it as a limitation.]
Response 1:
We thank the reviewer for this suggestion. We now have additional information in the edited manuscript (lines [110-112] [94-95], ).
[The optimal frame rate was achieved by reducing the sector width and scan depth. This was sufficient even for such a high heart rate. The shallow location and small size of the neonatal heart create favorable conditions for obtaining high-quality images. Tracking quality was assessed automatically by the software; if inadequate tracking was detected, additional manual correction was performed.]
line# 94 - 95 [Patients with poor tracking were excluded from the analysis.]
line# 557 - 558 [All images were acquired by a single operator, so interoperator variability was not assessed.]
Comments 2: [Specify any sedation policy or handling technique during imaging to minimize motion artifacts. Provide a brief justification for sample size (e.g., power estimate or precedent in similar studies).]
Response 2: We have added a discussion on this aspect in the [Material and Methods] section ( lines [95-96]) to provide more clarity.
Our center does not use sedation for newborn examinations. The baby was either observed while asleep or waited until calm.
This is a prospective study. It should be noted that the sample size is limited: 67 patients were included over a 1.5-year period (the study group was n=65). The deadline for data collection was determined by the end of project funding. Of course, to increase statistical power and the overall significance of the findings, such an analysis is preferably conducted on a cohort comprising several hundred or thousands of observations.
Regarding the validity of the chosen sample size, there are precedents in the modern scientific literature. In particular, a number of relevant studies have used cohorts of comparable size—approximately 70 observations—to address similar issues. For example, in the study by Kanagaraj et al.
Kanagaraj, U.K., Castaldo, M., Braschel, M. et al. Reliability of two-dimensional versus M-mode echocardiography for left atrium/aortic diameter ratio and fractional shortening in extremely preterm infants. Pediatr Res (2025). https://doi.org/10.1038/s41390-025-04389-z
Comments 3 : [The results are comprehensive but somewhat dense. Consider summarizing key quantitative findings (e.g., representative median strain values or base-to-apex gradients) in the main text rather than referring extensively to supplementary material.]
Response 3:Thank you for your comment. We have added the final results diagram in Fig.10
Comments 4 : [The discussion is thorough and well-informed. However, you could further emphasize the clinical applicability of your reference values—how might clinicians use these data to identify early myocardial dysfunction?]
Response 4: Thank you for your comment. We have added a discussion on this aspect in the [Introduction] section (lines [41-49]) and in the [Discussion] section (lines [538-540]) to provide more clarity.
[ The biplane Simpson method of assessing global LV function results in high inter- and intra-operator variability and low reproducibility. A more accurate method of assessing LV systolic function is global longitudinal strain, which enables minor changes to be identified at an earlier stage [3]. It is suggested that segmental strain testing may help in detecting focal lesions of the LV myocardium, such as those associated with neonatal coronary pathology (coronary artery origin anomalies, thrombosis, fibroelastosis, etc.). In the presence of segmental pathology, the function of the affected segment may be compensated for by the increased function of healthy adjacent segments, and global systolic dysfunction indices may be within normal limits [2].]
[We hypothesize that strain values outside the reference range may indicate regional myocardial pathology. Further investigation is required to confirm this hypothesis.]
Comments 5 : [Please discuss potential effects of physiological variability (e.g., PDA status, respiratory support type) on strain values, even if not statistically significant.]
Response 5: Thank you for your comment.
We can assume that high parameters of artificial ventilation reduce preload, which may lead to a decrease in deformation parameters.
In the presence of patent ductus arteriosus, at older ages, when the ductus becomes hemodynamically significant, LV overload occurs, which can alter deformation parameters (from improvement to deterioration). This hypothesis requires further study.
We have added a discussion on this aspect in the [Discussion] section (lines [474-477])
Round 2
Reviewer 3 Report
Comments and Suggestions for Authors
Thank you for the detailed responses. All my comments have been fully addressed in the revised manuscript.